

# Tumor-preventing activity of aspirin in multiple cancers based on bioinformatic analyses

Diangeng Li[1,*], Peng Wang[3,*], Yi Yu[1], Bing Huang[1], Xuelin Zhang[1], Chou Xu[1], Xian Zhao[1], Zhiwei Yin[4], Zheng He[5], Meiling Jin[2] and Changting Liu[1]

[1] Chinese PLA General Hospital, Nanlou Respiratory Diseases Department, Beijing, China
[2] Beijing Chao-yang Hospital, Department of Nephrology, Beijing, China
[3] Chinese PLA General Hospital, Nanlou Medical Oncology Department, Beijing, China
[4] Hebei Medical University, School of Chinese Integrative Medicine, Shijiazhuang, China
[5] Chinese PLA General Hospital, Department of Clinical Laboratory, Beijing, China
[*] These authors contributed equally to this work.

## ABSTRACT

**Background**. Acetylsalicylic acid was renamed aspirin in 1899, and it has been widely used for its multiple biological actions. Because of the diversity of the cellular processes and diseases that aspirin reportedly affects and benefits, uncertainty remains regarding its mechanism in different biological systems.

**Methods**. The Drugbank and STITCH databases were used to find direct protein targets (DPTs) of aspirin. The Mentha database was used to analyze protein–protein interactions (PPIs) to find DPT-associated genes. DAVID was used for the GO and KEGG enrichment analyses. The cBio Cancer Genomics Portal database was used to mine genetic alterations and networks of aspirin-associated genes in cancer.

**Results**. Eighteen direct protein targets (DPT) and 961 DPT-associated genes were identified for aspirin. This enrichment analysis resulted in eight identified KEGG pathways that were associated with cancers. Analysis using the cBio portal indicated that aspirin might have effects on multiple tumor suppressors, such as TP53, PTEN, and RB1 and that TP53 might play a central role in aspirin-associated genes.

**Discussion**. The results not only suggest that aspirin might have anti-tumor actions against multiple cancers but could also provide new directions for further research on aspirin using a bioinformatics analysis approach.

Corresponding authors
Meiling Jin, auml_1986@hotmail.com
Changting Liu,
changtingliu1212@sohu.com

## INTRODUCTION

Nonsteroidal anti-inflammatory drugs (NSAIDs) are efficacious preventive agents against several different types of malignancies, including colorectal cancer (*Bilani, Bahmad & Abou-Kheir, 2017*). Reports regarding risk reduction have shown impressive results with increasing NSAID intake showing a reduced relative risk of colon cancer by 63%, whereas it has shown a 39% reduction for prostate and breast cancer and 36% for lung cancer (*Harris et al., 2005*). A long-term observation of randomized, controlled trial cohorts with cardiovascular disease also revealed lower risks of developing colon malignancy and a

reduced incidence and development of metastatic disease, which are benefits that are attributed to regular aspirin use (*Gray et al., 2017*).

The recent advancements in biomedical research, such as multicenter genomic studies involving proteomics, microarrays and other high-throughput screening assays, has resulted in a staggering amount of candidate gene "hits"; more than enough to overwhelm subsequent thematic or phenotypic-based data analyses. Nevertheless, the network-based approach can be a simple and effective means of analyzing these gargantuan sets of data and permit researchers to uncover previously difficult to characterize genetic relationships between a drug, its targets and interacting proteins as well as its disease associations. It has been reported that establishing a drug target network can be accomplished using drug interaction databases (*Mestres et al., 2008*). There are several open-access databases for the collection of pharmacogenomics data. Drugbank is the most commonly used database. Drugbank's primary focus is compiling and curating information concerning drug targets (genetic and protein-specific data), drug metabolism, drug interactions, and the relationships between drugs and diseases or side effects (*Wishart, 2008*). However, Drugbank might not completely overlap with those in STITCH or the Therapeutic Target Database. In this study, we first identified direct protein targets (DPTs) using Drugbank and the STITCH database. We then identified proteins associated with these DPTs using the Mentha database. Finally, we built an aspirin-target network. Enrichment analysis was used to analyze the proteins of this network. This method of analysis permits a deeper understanding of how aspirin may prevent cancer and drive the development of future chemotherapeutic medication.

## MATERIALS AND METHODS

### Drug-target search

In this study, Drugbank (https://www.drugbank.ca/) (*Wishart et al., 2006*) and STITCH (http://stitch.embl.de/) (*Kuhn et al., 2007*) were utilized to identify aspirin-target interactions to produce an aspirin-target network. A visualization chart was constructed with the resultant data, followed by more extensive data analysis and proposals for subsequent validation experiments.

### Network generation/visualization and analysis of gene enrichment sets

Mentha (http://mentha.uniroma2.it/) was used to analyze protein–protein interactions (PPIs) to find DPT-associated genes with the 0.3 set as the minimum interaction scores (*Calderone, Castagnoli & Cesareni, 2013*). DAVID was used for the GO enrichment analysis and KEGG enrichment analysis (*Huang, Sherman & Lempicki, 2008*).

### Aspirin-linked cancer genomic data exploration using the cBio cancer genomics portal

The cBio Cancer Genomics Portal (http://cbioportal.org) represents a free platform that allows multidimensional exploration of cancer genomic data by translating molecular profiles sequenced from cell lines and cancer tissues into easily comprehensible proteomic,

gene expression, epigenetic and genetic events (*Cerami et al., 2012*). With the cBio Portal, we explored the connections of aspirin-associated genes across the genetic databases of several cancer-related studies. Using the portal search function, all of the aspirin-associated genes found in cancer study samples were categorized as altered or not altered. We were also able to construct multiple visualization platforms by grouping the cancer data alterations based on aspirin gene data sets.

## RESULTS

### Identification of DPTs

Drugbank and STITCH were used to identify direct protein targets (DPTs) of aspirin; these 18 primary DPTs of aspirin were PTGS1, PTGS2, AKR1C1, PRKAA1, EDNRA, IKBKB, TP53, HSPA5, RPS6KA3, NFKBIA, NFKB2, CRP, SELP, TBXA2R, REN, MMP9, NOS3, and IL10. Then, we used Mentha to analyze protein–protein interactions (PPIs) to find DPT-associated genes and uncovered 961 unique target-protein interactions, which we determined to be aspirin-related DPT-associated genes along with the 18 primary targets (Table 1).

### GO pathway analysis

The online DAVID software was used to determine overrepresented GO categories based on our previously identified DPT-associated genes. GO analysis revealed significant genetic enrichment in the area of biological processes (BP), which was comprised of the regulation of nucleobase, nucleoside, nucleotide and nucleic acid metabolism (27.4%), signal transduction (27.2%), cell communication (23.4%), protein metabolism (16.7%), apoptosis (3.9%), regulation of gene expression and epigenetics (1.5%), regulation of the cell cycle (1.5%), DNA repair (1.4%), regulation of cell growth (0.8%), and regulation of cell proliferation (0.8%). For the area of the cell components (CC), these genes were enriched in the nucleus (70.6%), cytoplasm (64.3%), nucleolus (21.5%), cytosol (21%), exosomes (20.1%), nucleoplasm (13.2%), centrosome (11.8%), ribonucleoprotein complex (2.4%), protein complex (2.1%) and PML body (1.9%). Additionally, GO molecular function (MF) analyses showed that these genes were significantly enriched in transcription regulator activity (11.3%), ubiquitin-specific protease activity (9.9%), transcription factor activity (9.7%), protein serine/threonine kinase activity (8.7%), receptor signaling complex scaffold activity (5.3%), chaperone activity (2.5%), protein binding (2.3%), protein serine/threonine phosphatase activity (1.1%), DNA repair protein (1.1%), and DNA topoisomerase activity (0.4%) (Fig. 1).

### KEGG pathway analysis

The functional characteristics of these aspirin-related genes were characterized by the use of the KEGG pathway enrichment analysis, which is a feature embedded in the DAVID software. The top 10 KEGG pathways linked to aspirin DPTs and their DPT-associated genes include Epstein-Barr virus infection (63 genes), ubiquitin-mediated proteolysis (46 genes), pathways in cancer (78 genes), cell cycle (40 genes), NF-kappaB signaling pathway (33 genes), herpes simplex infection (47 genes), TNF signaling pathway (35

**Table 1  Identification of DPT-associated genes using mentha.**

| # | DPT of aspirin | DPT-associated genes |
|---|---|---|
| 1 | PTGS1 | PTGS2, CAV2, CAV1, PTGIS, NCL |
| 2 | PTGS2 | EP300, USP22, COPS7A, ELAVL1, CAV1, ELMO1, COPS5, DERL1, NUCB1, PTGIS, CTNNB1, TP53, PTGS1, APP, CELF2, AGTR1, NOS2 |
| 3 | AKR1C1 | COMMD8, TFF3, PTPN3, MAPK3 |
| 4 | PRKAA1 | RIMBP3, TRIP6, L3MBTL3, ABI1, ARHGAP22, HMBOX1, NRBF2, MTFR2, PRKAG2, VPS37B, STK11, CDC37, FKBP5, HOMEZ, RBPMS, THAP1, PRKAB1, IKZF3, MAGED1, ROPN1, PHC2, SDE2, PNMA5, CHERP, VPS52, CAMKK1, RFX6, INO80E, EMILIN1, THAP7, PPM1E, MORC4, LZTS2, XRN2, PLEKHA4, MTUS2, UBXN11, CRTC2, TFPT, WDR62, NAB2, SORBS1, RC3H1, PRKAG1, KRT40, RPTOR, FNIP1, CRBN, TXNIP, ABI2, HDAC5, SSX2IP, CTBP1, TOMM34, USP10, TSC22D4, TRIM27, MAP3K7, PPM1F, FANCA, BHLHE40, ZBED1, APRT, CFTR, ACACA, ULK1, CDX4, HSP90AB1, PRKAB2, KIF1C, FANCG, HSP90AA1, FSBP, HSPB1, CPE, PPP2CA, NEDD1, PSMD11, MDM4, EPM2A, TP53, MTOR, PRKAA2, AES, GOLGA2, PPM1A, TSC2 |
| 5 | EDNRA | ARRB1, ARRB2, HDAC7, KAT5, COPS5, EDN1, SCR |
| 6 | IKBKB | PRKCA, CREBBP, IRAK1, ERC1, CTNNB1, FANCA, PPP2CA, TP53, AURKA, COPS4, MUC1, TNFAIP3, TNFRSF1A, AKT1, PRKCD, MAP3K1, MAVS, TNF, PRKCE, TRIM21, RIPK1, RICTOR, RELA, TRAF3IP2, NCOA3, COPS3, STAP2, MAP3K7, NFKB1, MAP3K14, TRAF1, TRAF2, CDC37, NFKBIB, KEAP1, NFKBIA, CHUK, IKBKG, EGLN3, COPS7A, USP18, LATS2, HOMER3, BTRC, PPM1B, BCL10, TAX, JUN, HSP90AB1, SRC, TRIM27, PEBP1, MTOR, HTT, MEOX2, CSNK2B, HNRNPU, PRKCQ, IKBKE, TWIST1, MAPK14, TRIM40, BRAP, NLRC5, RPTOR, TNIP2, PELI1, NEDD4L, TRAPPC9, TP63, CUEDC2, TAB2, KLHL21, FAF1, TRAF6, FOXO3, TP73, HSP90AA1, GLI1, PPP2R3C, CSF2RA, TSC1, COPS5, NAA20, PPP1CA, PRKDC, CSF2RB, CLTC, PPARG, PLK1, ROCK1, MAP3K11, PRKCI, ORF71, SNAP23 |
| 7 | TP53 | MT1A, CDK1, S100A6, BRCC3, RPL5, BANP, BRCA2, UFD1L, GPX2, CDKN1A, HSP90B1, P0DMV9, P0DMV8, RCC1, NQO1, XRCC1, CREB1, HNF4A, MTOR, TP53, BMI1, PPIF, NMT1, PHB, ZBTB2, SFN, YWHAZ, VRK2, SET, HECW1, Q7L7W2, BRE, RBBP6, HSPA4, UBB, TOP2A, VDR, EEF2, TPT1, HSC82, TNFAIP3, MDM2, NFKBIA, MAPK3, |
| 8 | HSPA5 | AMFR, PPP2R2B, MAPRE1, SIL1, ERLEC1, CFTR, Q6T424, SEC61A1, CBL, SNW1, MAP1LC3A, DMKN, AGO4, SPG20, OS9, EIF2AK3, DNAJB11, HNRNPA3, PDIA6, MTNR1A, MTNR1B, HSPA8, DNAJC10, PAWR, SH3BP4, SEC63, UBQLN4, HSPBP1, AKT1, YWHAB, CPT1A, GRB2, RELA, EP300, SQSTM1, GPX7, DPH1, HDAC6, UCHL5, DNAJC1, UBL4A, UBE3A, ID2, DNAJC3, FUS, TMEM132A, VHL, CLU, P01266, VIM, RPN1, AIRE, ERLIN2, TP53, RAF1, EGFR, F7VJQ1, SVIL, PRNP, ERN1, HNRNPA1, FCHSD2, A2M, LDLR, HSPB1, KRT8, PIAS1, P0DMV8 |
| 9 | RPS6KA3 | BARX1, EIF3C, CREBBP, CSNK2B, TRAF2, YBX1, SMS, HIST1H3J, PEA15, FBXO43, MAPT, FGFR1, ATP5J, MAPK1, PDPK1, MAPK3, NFKBIA, HMGN1, H2AFX, MASP1 |
| 10 | NFKBIA | MTOR, ZNF212, NFKBIB, ARRB2, UBA52, RPS6KA1, COMMD1, DNAJA3, UBE2D3, PTPN13, NEDD9, CUL1, SUMO4, ARRB1, VCP, REL, UL54, FBXW11, SKP1, RELB, IKBKG, BTRC, RELA, TBK1, G3BP2, HIF1AN, RWDD3, PIK3R1, CSNK1A1, UBE2D1, COPS2, UBE2I, CSNK2B, MAPK14, MAP3K14, BARD1, IKZF4, NCOR2, POM121, UBE2D2, DYNLL1, UBE2L3, NFKB2, MAP3K1, ITPK1, IKBKE, RPS6KA3, CDC34, ABL1, TP53, RPS3, ATF4, UBC, IKBKB, CHUK, NFKB1, POLRMT, SOCS3, MAP3K7, PRKCA, PSMA4, PIR, AURKA, IKBKAP, BCL10, CAPN1, HOXB7, HNRNPA1, SLC25A4, PSMA2, LYL1, TNF |
| 11 | NFKB2 | NFKB2, REL, TSC22D3, FBXW7, FBXW11, RELA, RELB, BTRC, MAP3K14, DPF2, EPS8, NFKBIB, STAT3, BCL3, NFKB1, CHUK, MAP3K8, MEN1, NKRF, NR3C1, SP1, P0C722, P0C723, P03207, NFKBIE, NFKBIA |
| 12 | CRP | RPL13A, DGCR14, MAPK3, LAMC3, CRP, APCS, C1QA, CFH, MAPK1, FCGR2C, HIST1H1A, HIST2H2AC, GMPPA, FN1, SPA, FCN2, GLUD1 |
| 13 | SELP | SELP, CD24, COL18A1, SELPLG, SNX17, SNX27, AP1M1, VCAN, GP1BA, EZR |
| 14 | TBXA2R | GNA11, GNB1, KCNMA1, PSME3, NME2, AAMP, YWHAZ, GNB2L1, RPGRIP1L, PKN1, RAB11A, PRDX4, GNAQ, GHRL, GNG2, GNAS, KCNMB1, GPRASP1, WDR36, GRK5, PRKG1, GNA13, GRK6, PTGIR, PRKCA, PSMA7, SIVA1, PRKACA, GNAI2 |
| 15 | REN | AGT, ATP6AP2 |
| 16 | MMP9 | COL18A1, CXCL5, CXCL6, VCAN, COL1A2, CLU, TIMP1, TGFB1, SRGN |
| 17 | NOS3 | ACTN2, CTNNB1, MPRIP, IMMT, EFEMP1, H3F3B, TXNDC11, HTRA1, RNF31, FIS1, CAV1, NOSTRIN, AKT1, NOSIP, ST13, GCDH, ACTB, GUCY1B3, GOLGA2, CDC37, MAGEA11, ACTN4, EFEMP2, UMPS, P0DP23, P0DP24, P0DP25, HSP90AA1, TAX, APOE, P0DP30, P0DP31, P0DP29 |
| 18 | IL10 | IL10RA, IL10RB, A2M |
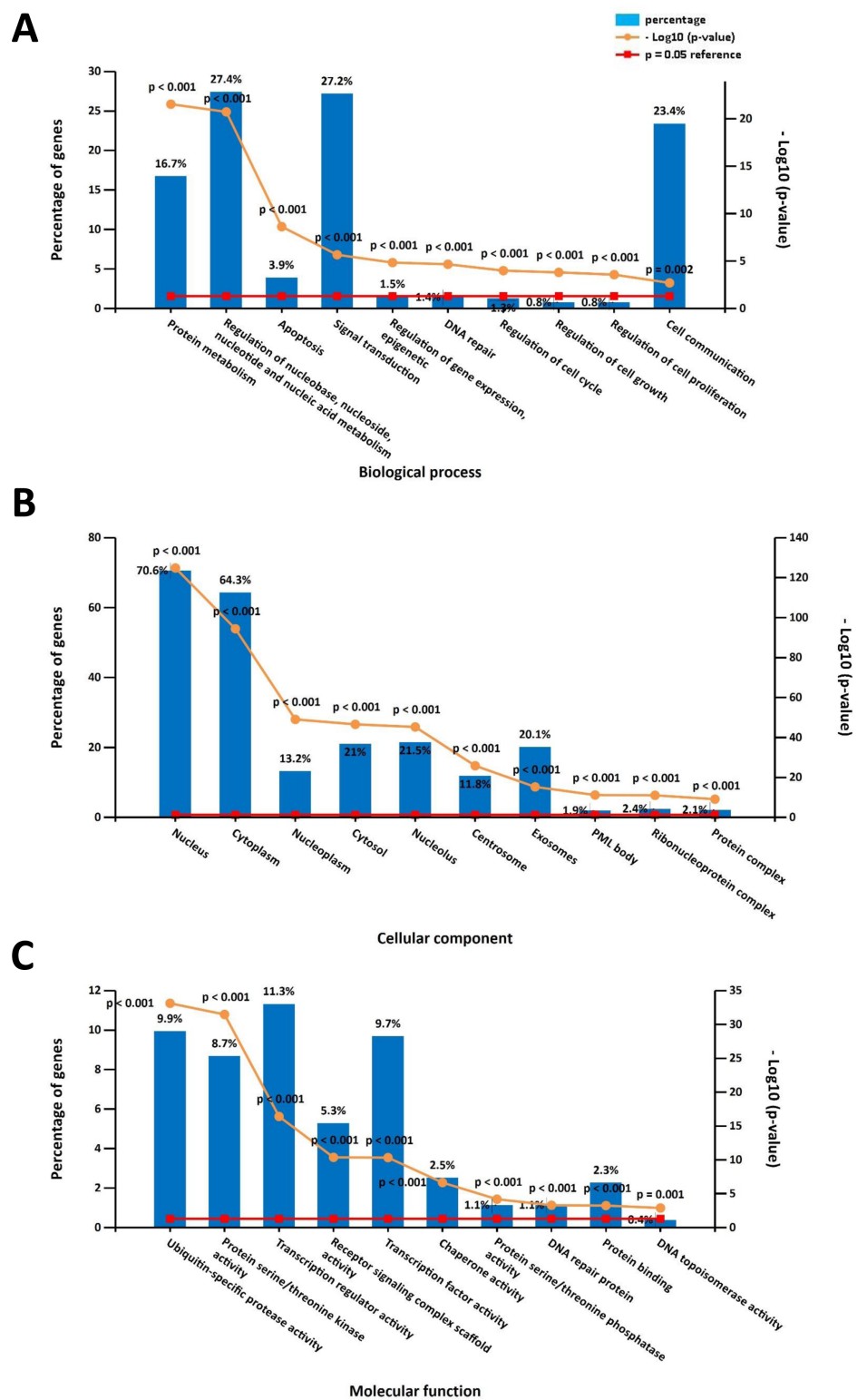

**Figure 1** Gene ontology (GO) enrichment analysis on the direct protein targets (DPTs) and DPT-associated genes of aspirin. (A) GO biological process (BP) analysis, (B) GO cell component (CC) analysis, (C) GO molecular function (MF) analysis.

**Table 2    KEGG pathway associated with cancer.**

| KEGG pathway-term | Count | $p$ Value | Genes |
|---|---|---|---|
| bta05200: Pathways in cancer | 78 | 2.79E−20 | GNA13, HSP90AB1, PTGS2, MMP9, GNA11, PPARG, NFKB1, NFKB2, PTEN, TGFB1, CTNNB1, GLI1, AKT1, EDNRA, AGTR1, CDKN2A, CASP8, PRKACA, GNG2, NOS2, CHUK, PRKCA, CTBP1, HSP90AA1, BCR, ROCK1, RELA, TP53, RB1, DAPK3, CDK2, RAD51, DAPK1, MAPK1, HIF1A, GNAQ, GNB1, LAMC3, JUN, MAPK3, MAPK9, MDM2, PIAS2, GNAS, MAPK8, TRAF1, TRAF2, FGFR1, GNAI2, PML, EGLN3, NFKBIA, BCL2L1, PTK2, BCL2, TRAF6, PIK3R1, FN1, AXIN1, DVL2, MSH2, VHL, CREBBP, BRCA2, SMAD3, SMAD2, STAT3, HSP90B1, LAMA4, CDKN1A, HDAC2, HDAC1, GSK3B, IKBKG, PLCG2, MTOR, IKBKB, ABL1 |
| bta05215:Prostate cancer | 28 | 6.53E−13 | HSP90AB1, FGFR1, NFKBIA, NFKB1, PTEN, CTNNB1, AKT1, PDPK1, BCL2, CHUK, PIK3R1, HSP90AA1, RELA, CREB1, CREBBP, TP53, RB1, CDK2, MAPK1, CDKN1A, ATF4, HSP90B1, GSK3B, MAPK3, IKBKG, MDM2, MTOR, IKBKB |
| bta05212:Pancreatic cancer | 21 | 1.02E−09 | RELA, TP53, SMAD3, BRCA2, SMAD2, NFKB1, BCL2L1, RB1, STAT3, TGFB1, RAD51, AKT1, MAPK1, CDKN2A, MAPK3, IKBKG, MAPK9, MAPK8, IKBKB, CHUK, PIK3R1 |
| bta05222:Small cell lung cancer | 24 | 2.04E−09 | TRAF1, TRAF2, PTGS2, RELA, TP53, NFKBIA, NFKB1, BCL2L1, RB1, PTEN, CDK2, AKT1, LAMA4, PTK2, LAMC3, BCL2, IKBKG, PIAS2, NOS2, TRAF6, IKBKB, CHUK, PIK3R1, FN1 |
| bta05210:Colorectal cancer | 16 | 8.94E−06 | MSH2, TP53, SMAD3, SMAD2, TGFB1, CTNNB1, AKT1, MAPK1, GSK3B, JUN, BCL2, MAPK3, MAPK9, MAPK8, PIK3R1, AXIN1 |
| bta05219:Bladder cancer | 11 | 1.21E−04 | MAPK1, CDKN1A, CDKN2A, MMP9, MAPK3, TP53, MDM2, RB1, DAPK3, SRC, DAPK1 |
| bta05213:Endometrial cancer | 11 | 9.87E−04 | AKT1, MAPK1, PDPK1, GSK3B, MAPK3, TP53, FOXO3, PTEN, PIK3R1, AXIN1, CTNNB1 |
| bta05223:Non-small cell lung cancer | 11 | 0.002081 | AKT1, PRKCA, MAPK1, PDPK1, CDKN2A, MAPK3, PLCG2, TP53, RB1, FOXO3, PIK3R1 |
| bta05211:Renal cell carcinoma | 11 | 0.007081 | AKT1, MAPK1, HIF1A, VHL, JUN, MAPK3, CREBBP, EGLN3, RAP1B, TGFB1, PIK3R1 |

genes), toxoplasmosis (37 genes), viral carcinogenesis (52 genes), and FOXO signaling pathway (37 genes). We primarily focused on the KEGG pathways associated with cancers: prostate cancer (28 genes), pancreatic cancer (21 genes), small-cell lung cancer (24 genes), colorectal cancer (16 genes), bladder cancer (11 genes), endometrial cancer (11 genes), non-small-cell lung cancer (11 genes), and renal cell carcinoma (11 genes; Table 2).

## Mining genetic alterations and networks of aspirin-associated genes in cancer with the cBio portal
### Prostate cancer

There were large variations of 24.23% to 73.3% in the gene sets analyzed among 9 prostate cancer gene analysis studies. OncoPrint results showed that 1412 (50%) cases had an alteration in at least one of these 28 gene sets (PTEN 18%, TP53 16%, RB1 8%, IKBKB 7%, HDAC2 7%, FGFR1 6%, PIK3R1 5%) (Fig. 2A and Fig. S1). With the help of the CBio portal, we were able to obtain interactive analyses and view constructed networks of genes that were altered in cancer. Figure 3A depicts a gene network consisting of PTEN, TP53, and IKBKB genes and their respective gene neighbors. PTEN and TP53 may play important roles in this network.

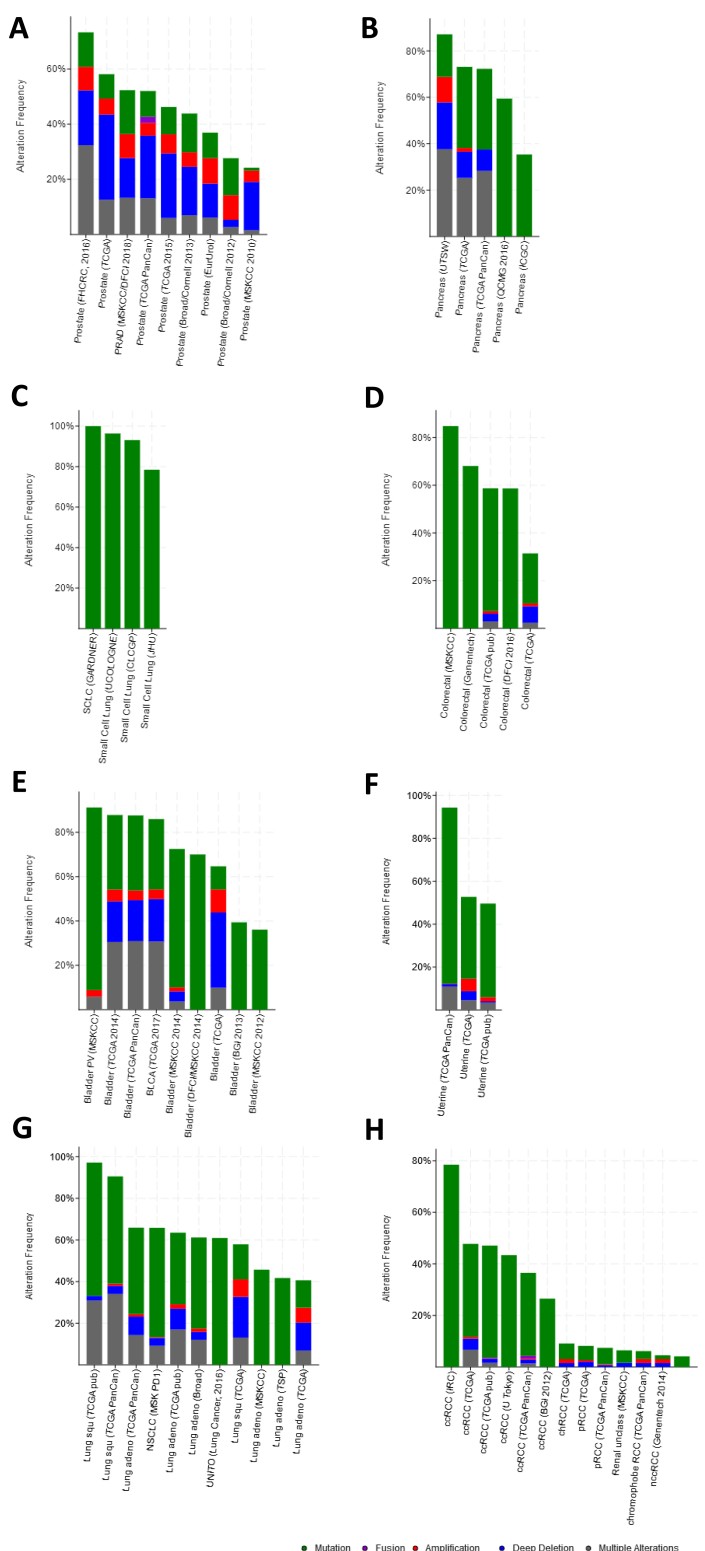

**Figure 2 Mining genetic alterations connected with aspirin-associated genes in cancer studies with the cBio cancer genomics portal.** (A) Prostate cancer, (B) pancreatic cancer, (C) small-cell lung cancer, (D) colorectal cancer, (E) bladder cancer, (F) endometrial cancer, (G) non-small-cell lung cancer, (H) renal cell carcinoma.

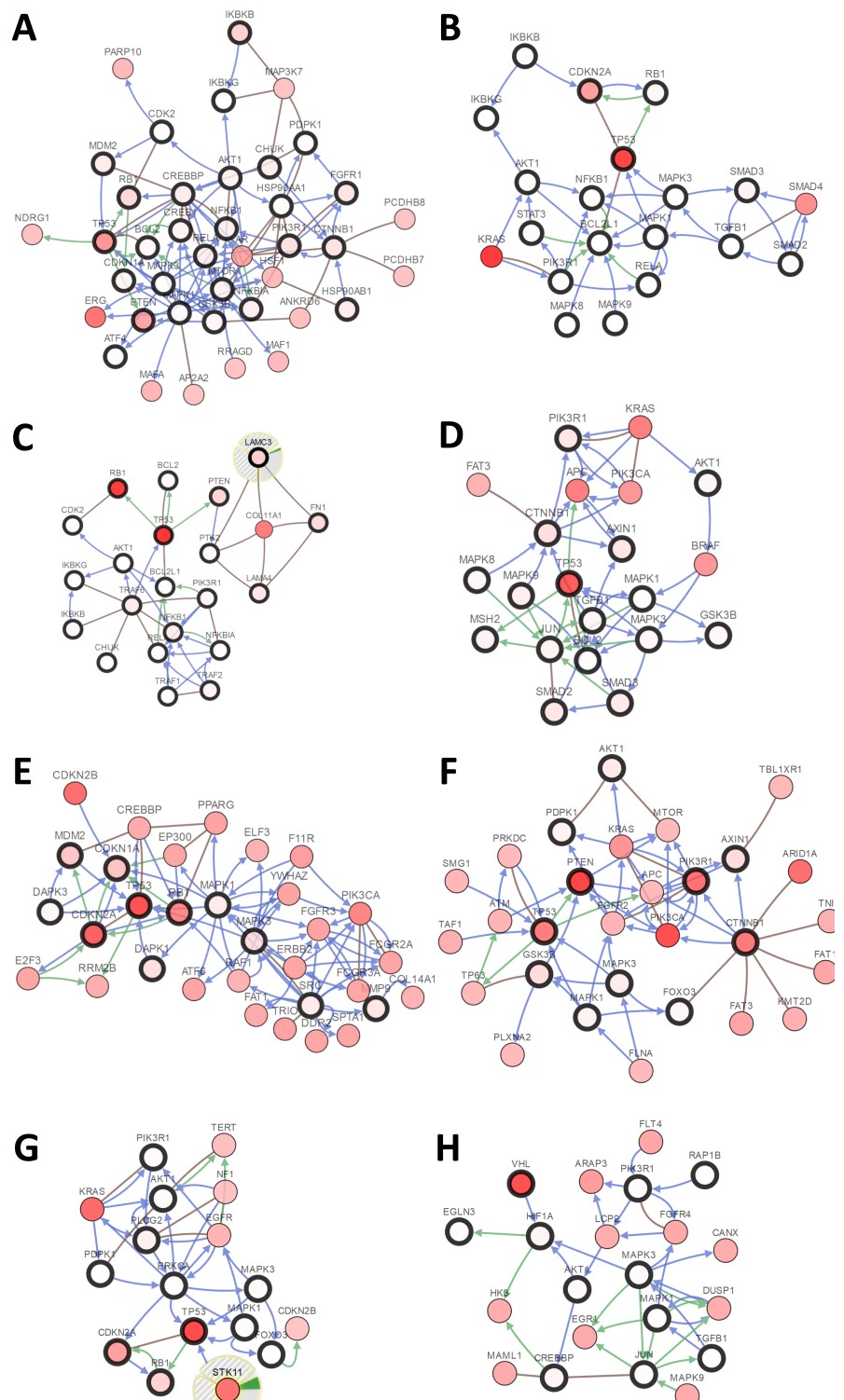

**Figure 3** **Visual display of the gene network connected to genes in cancer.** (A) Prostate cancer,
(B) pancreatic cancer, (C) small-cell lung cancer, (D) colorectal cancer, (E) bladder cancer,
(F) endometrial cancer, (G) non-small-cell lung cancer, (H) renal cell carcinoma.

### Pancreasubtic cancer

The gene sets for the five analyzed pancreatic cancer studies revealed variations of 35.35% to 87.16% among the gene sets. The results showed that 670 (70%) cases had an alteration in at least one of these gene sets (TP53 58%, SMAD2 4%, TGFB1 2.5%, AKT1 2.3%, RB1 1.9%, SMAD3 1.7%, and STAT3 1.7%) (Fig. 2B and Fig. S2). As shown in Fig. 3B, TP53 may play an important role in this network.

### Small-cell lung cancer

Upon the analysis of four small-cell lung cancer studies, we noted alterations of 78.43% to 100% between the gene sets. The OncoPrint results showed that 193 (91.9%) cases had an alteration in at least one of the 24 gene sets (TP53 86%, RB1 65%, FN1 12%, PTEN 8%, LAMC3 5%, NOS2 4%, and LAMA4 3%) (Fig. 2C and Fig. S3). As shown in Fig. 3C, there was a close relationship between TP53 and RB1, and TP53 may play an important role in this network.

### Colorectal cancer

There were variations of 31.41% to 84.78% for the five colorectal cancer study gene sets that we interpreted. The results showed that 892 (51.1%) cases had an alteration in at least one of these gene sets (TP53 37%, SMAD2 5%, CTNNB1 4%, PIK3R1 4%, and SMAD3 3%) (Fig. 2D and Fig. S4). We focused primarily on TP53, and the network of TP53 is shown in Fig. 3D.

### Bladder cancer

We observed alterations ranging from 36.08% to 91.18% in gene sets from the nine analyzed bladder cancer studies. The results show that 1316 (74.9%) cases had an alteration in at least one of the 11 gene sets (TP53 41%, CDKN2A 31%, RB1 20%, CDKN1A 9%, and MDM2 8%) (Fig. 2E and Fig. S5). The network of these genes is shown in Fig. 3E, and TP53 and CDKN2A may play an important role in this network.

### Endometrial cancer

The three endometrial genetic studies that we analyzed had gene set variations ranging from 49.6% to 94.33%. The results showed that 1036 (71%) cases had an alteration in at least one of these gene sets (PTEN 48%, PIK3R1 24%, TP53 23%, CTNNB1 20%, and AXIN1 5%) (Fig. 2F and Fig. S6). The network of these genes is shown in Fig. 3F.

### Non-small-cell lung cancer

Among the analyzed NSCL cancer studies, alterations ranging from 40.61% to 97.19% were found for the submitted gene sets. The results showed that 2046 (64%) cases had an alteration in at least one of these gene sets (TP53 49%, CDKN2A 25%, EGFR 17%, PIK3CA 9%, and RB1 7%) (Fig. 2G and Fig. S7). The network of these genes is shown in Fig. 3G. This indicates that TP53 may play important roles in the occurrence of NSCLC.

### Renal cell carcinoma

The renal cell carcinoma studies included in our analysis displayed intergene set alterations of 4.11% to 78.48%. The results showed that 827 (30%) cases had an alteration in at least

one of these gene sets (VHL 27%, CREBBP 1.4%, and AKT1 0.6%) (Fig. 2H and Fig. S8). The network of these genes is shown in Fig. 3H.

## DISCUSSION

Acetylsalicylic acid was renamed aspirin in 1899 (*Fuster & Sweeny, 2011*). In 1988, a case-control study was the first to record a negative correlation between colorectal cancer and aspirin use (*Kume et al., 2010*), which suggests that aspirin might be protective against cancer. Further investigations based on cohorts of cardiovascular disease patients taking aspirin found that aspirin may generally lower the risk of cancer. Six separate trials that analyzed patients who took daily low-dose aspirin (75 mg and above) for three years revealed that aspirin conferred an overall relative risk of 0.76 for cancer with a longer duration of aspirin intake resulting in higher benefits (*Rothwell et al., 2012*). In fact, several lines of evidence highlight that aspirin may be beneficial in decreasing mortality in cancer, especially colorectal cancer-related death. This protection may also extend to other malignancies, such as prostate, lung, breast and gastroesophageal cancers. Given the strong epidemiological evidence, it is hypothesized that aspirin may act on common cancer pathways to suppress cancer progression and metastases (*Cao et al., 2016*). In 2007, the United States Preventive Services Task Force (USPSTF) initially discouraged aspirin use for preventing colorectal cancer. However, the updated USPSTF 2015 recommendations acknowledge the existence of several compelling sources of evidence and included colorectal cancer prevention into the rationale for routine, low-dose aspirin intake for those with specific cardiovascular risk profiles between the ages of 50 to 69. This landmark decision was the first to endorse a pharmacological compound for use as a preventive agent against cancer in a population not specifically known to have a high risk of developing malignancies. Despite these advancements, we still possess a limited understanding of how aspirin exerts its benefits. Our study utilized bioinformatics methods to establish a drug target network to dissect the underlying molecular mechanisms of aspirin in cancer. We first determined primary aspirin DPTs and functionally linked them to their respective proteins with the help of drug interaction databases and protein–protein interaction database (Drugbank, STITCH, and Mentha). Next, using samples from large-scale cancer genomic projects in the cBio portal, we verified if there were previously identified genetic alterations that were characterized for aspirin-associated genes/proteins. This method allowed us to clearly map out aspirin-related DPTs and their associated genes to their biological pathways using the available databases. Not only does this information contribute to the current knowledge of how aspirin prevents cancer, it also uncovers potential treatment targets and provides new directions for cancer therapeutics.

Using the tools available on the online platform, we identified 18 primary DPTs, 961 secondary DPT-associated genes/proteins, and eight enriched KEGG pathways linked to aspirin-associated genes. These eight enriched KEGG pathways included several cancers. The cBio portal was used to analyze associations between these genes and cancer based on the TCGA database. The results show that most of the gene protein targets could be found to have alteration in cancer samples, and the network analysis showed that TP53,

PTEN, and RB1 might play important roles in the mechanism of aspirin. Human cancers commonly display mutated or inactivated versions of the TP53 and PTEN tumor suppressor genes. TP53 is a crucial cell cycle regulator and is responsible for inducing apoptosis. As shown in Fig. 3, TP53 was found to possess a central role in the gene networks that we constructed. A large proportion of genetic defects in prostate cancer were identified to be mutations or deletions that result in attenuations of TP53 and PTEN expressions and culminate in enhanced carcinogenesis. By controlling PTEN transcription, p53 can suppress tumorigenesis when there is PTEN deficiency. It has been reported that copy number alterations of p53 and RB1 could be prognostic markers in prostate cancer as RB1 and TP53 were found to cooperate in suppressing metastasis (*Ku et al., 2017*). Functionally inactivating RB1 and TP53 appeared to be enough to stimulate SCLC development in mice, whereas restoring their expression in human SCLC cell lines halted further tumorigenesis by the induction of G1-arrest and cell apoptosis (*Fiorentino et al., 2016*). It has been reported that mutations in TP53 and CKDN2A define the genetic landscape of pancreatic ductal adenocarcinoma. Alterations in TP53 can promote invasion and metastasis by increasing PDGFRB transcription and reversing the repressive function of the p73/NF-Y complex (*Weissmueller et al., 2014*). The p16 protein is encoded by the CDKN2A gene that resides on chromosome 9p21 and operates as a tumor suppressing gene. It represents a crucial cyclin-inhibiting cell cycle mediator, which serves to protect against premature cell transition from the G1 into the S phase. It was reported that a higher proportion of mutations occurred in CDKN2A in sample probands with familial pancreatic cancer (*Zhen et al., 2015*). NF-κB is upregulated in prostate cancer, whereas the knockdown of NF-κB decreased the expression of survivin, which is an important anti-apoptotic protein and NF-κB target gene, and induced capase-3 cleavage (*Zhuang et al., 2014*). Thus, IKBKB was named after its function of phosphorylating I$\kappa$B molecules, which is the inhibitor of NF-κB transcription factors (*Schmid & Birbach, 2008*), and indicates that IKBKB could act as a tumor suppressor. The Forkhead Box O family of transcription factors is comprised of three principal members, FOXO1, FOXO3, and FOXO4, which facilitate intracellular processes, such as glucose metabolism, cell differentiation, cell cycle regulation and other cellular functions. As a tumor suppressor, FOXO1 negatively regulates the highly oncogenic phosphatidylinositol 3-kinase (P13K)/AKT signaling pathway (*Wallis et al., 2015*). For colorectal cancer, aspirin has been recommended for use in the prevention of CRC. The PIK3CA mutation has been found to be a potential predictive biomarker for CRC (*Ogino et al., 2014*). Among the significantly enriched pathways from the KEGG analysis, many pathways have been proven to be involved in cancer metastasis, such as the FoxO signaling pathway (*Lin et al., 2015*), the AMPK signaling pathway (*Goodwin et al., 2014*), and the MAPK signaling pathway (*Li et al., 2016*). This evidence suggests that aspirin might also take part in the process of cancer metastasis and this should be verified in the further research. It is noteworthy that apart from the cancers identified in this study, aspirin might also have chemoprotective activity on other cancers, such as melanoma and ovarian cancer. Previous studies suggest that long-term aspirin use may be associated with a reduced risk of melanoma, especially among women (*Famenini & Young, 2014*; *Gamba et al., 2013*). Aspirin use was also associated with a reduced risk of ovarian cancer, especially among

daily users of low-dose aspirin (*Trabert et al., 2014*). If there was a continuous annotation update in the database, then more targets would be found in aspirin.

Thus, aspirin has anti-tumorigenic and chemopreventative activities in multiple tumors based on evidence from the bioinformatics analysis. In this study, the bioinformatics analysis helped visualize the molecular network bridging connectivity between aspirin-associated genes, aspirin and its primary targets, which demonstrates that these components are functionally related. This phenomenon may be biologically linked to the clinical impact that aspirin has on cancers, which may facilitate understanding of the tumor-preventing mechanism(s) of aspirin. Then, the molecular pathological epidemiology (MPE) could be used to study the "hot" proteins/genes as biomarker and individualized treatment as well as the outcomes. Although several limitations exist in this study, such as the verification of aspirin PPI, the evidence of a drug enrichment analysis baseline, and a lack of verification of clinical outcomes, all of these limitations will be the focus of further research. By establishing an aspirin target network, examining phenotypic variations in the context of aspirin-associated genes, and by characterizing cancer-specific gene signatures we gained insight into the role of aspirin in the prevention and treatment of diseases, including cancers.

## CONCLUSIONS

This bioinformatics analysis approach may significantly advance drug-disease research and increase our knowledge of the pathophysiology of malignant disease, which will significantly enhance our ability to devise techniques that can diagnose cancer earlier and more accurately. Given the rapid growth spurt in the field of aspirin biology, we hope that the results of this study will be able to provide new research directions for aspirin in cancer and for other human diseases.

### Funding
This work is supported by the China Postdoctoral Science Foundation (Grant numbers 2016M602971 and 2017T100809). The funders had no role in study design, data collection and analysis, decision to publish, or preparation of the manuscript.

### Grant Disclosures
The following grant information was disclosed by the authors:
The China Postdoctoral Science Foundation: 2016M602971, 2017T100809.

### Competing Interests
The authors declare there are no competing interests.

### Author Contributions
- Diangeng Li conceived and designed the experiments, performed the experiments, analyzed the data, contributed reagents/materials/analysis tools, prepared figures and/or tables.

- Peng Wang, Yi Yu and Xian Zhao performed the experiments, analyzed the data.
- Bing Huang and Xuelin Zhang performed the experiments, analyzed the data, contributed reagents/materials/analysis tools.
- Chou Xu performed the experiments, contributed reagents/materials/analysis tools.
- Zhiwei Yin and Zheng He analyzed the data.
- Meiling Jin conceived and designed the experiments, analyzed the data, authored or reviewed drafts of the paper, approved the final draft.
- Changting Liu conceived and designed the experiments, authored or reviewed drafts of the paper, approved the final draft.

## Data Availability

The raw data are provided in the Tables.

## Supplemental Information

Supplemental information for this article can be found online at http://dx.doi.org/10.7717/peerj.5667#supplemental-information.

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
