# Peer review of "Tumor-preventing activity of aspirin in multiple cancers based on bioinformatic analyses"

_PeerJ, doi:10.7717/peerj.5667_

## Round 0.1 · original submission · Major Revisions

As you can see, reviewers gave rather diversified recommendations, from accept to reject. I strongly encourage you to pay very close attention to the critical points raised by Reviewer #3, who is the well-established specialist in the field of computational biology and bioinformatics.

Reviewer 1 ·

Basic reporting

fine as is.

Experimental design

Data sources are heterogenous and might have created bias. The authors should address this.

Validity of the findings

Data sources are heterogenous and might have created bias. The authors should address this.

Additional comments

I have enjoyed reading this paper. This is overall a well conducted bioinformatic study. This is very timely topic as we now characterize medications and tumors deeply to personalize prevention, treatment and management. There are a few issues.


The authors should discuss molecular pathological epidemiology (MPE) research. Using human populations, MPE can deeply study phenotypes of disease outcome such as cancer using molecular pathologic analyses, and link factors such as environment, diets, and aspirin to molecular pathology, augment causal relationships. It would be great to have detailed discussion on application of tumor biomarkers and cancer subtyping to MPE, and its impact on medicine and populations. You can see reviews on MPE in Gut 2011, Epidemiology 2016, Gut 2018, etc.

Reviewer 2 ·

Basic reporting

concise , well structured article.

Experimental design

rigorous investigation was performed using various bioinformatics tools.

Validity of the findings

no comments

Reviewer 3 ·

Basic reporting

Article lacks an overview of relevant literature in the content of similar types of bioinformatics studies.

The discussion section is a long listing of results and a number of statements borrowed from literature that lack focus and a convincing analysis. I was lost as to what you are trying to prove and what you actually accomplished.

Minor issues
The abstract is loaded with jargon, especially in the methods part. This should be rewritten to be more accessible.
The writing is repetitive. The earlier sections of results are repeated in the discussion. This should be avoided.

Experimental design

The approach taken by the authors seems relatively naïve and the reported results seem rather preliminary in nature. There are a number of unexplained decisions that lead to likely incomplete/inadequate treatment of the topic and incompleteness/bias in the results including (but not limited to):
- Why did you use only DrugBank to annotate drug targets? – There are many other resources that should be use together with this database to annotate a more complete list of targets.
- Why did you use STRINGS, instead of a more reliable source, like mentha? How do you justify the 0.5 as the minimum score?
- How do you justify the assumption that PPIs that involve the aspirin targets would actually affect their partner proteins?

Validity of the findings

The results (enriched pathways, GO terms etc.) lack proper statistical analysis. The fact that a given term is enriched does not mean that this results in meaningful. This should be compared with a baseline to evaluate whether a similar results would be established for other drugs that are not recommended for treatment of cancer. Drug targets are a biased group of proteins and they are unlike a random set of proteins that you assume in your analysis.

The authors claim that “This method allows us to clearly map out aspirin-related DPTs and their associated genes to their biological pathways and clinical outcomes. Not only does this information contribute towards current knowledge of how aspirin prevents cancer, it also uncovers potential treatment targets, providing new directions for cancer therapeutics.” IMHO, this is not supported by the submitted manuscript. The authors generate mostly putative protein targets, few of them are in fact direct (given the use of STRINGS), they do not provide links to clinical outcomes – just speculate about such links, and finally I do not see how this provides new directions for therapeutics given that not a single new example of that was demonstrated by the authors.

Additional comments

This is a rather preliminary study that would require more work before future re-submission. The computational setup lacks depth and justification, statistical analysis is missing, and the authors made strong claims that in my view lack evidence.

Reviewer 4 ·

Basic reporting

1. The manuscript is written in clear and unambiguous, professional English.
2. The literature cited is satisfactory.
3. The article structure , figures and tables are professionally drafted.
4. The results and hypotheses projected in the study are satisfactory and self contained.

Experimental design

1. The bioinformatic analyses described in the study has potential for further experiments to validate potential targets of Asprin action.
2. the research question is well designed. The authors have taken a bioinformatic approach to mine the existing data to hypothesize the role of asprin in cancer biology.
3. The methods described are sufficient in context of the present study.

Validity of the findings

1, the findings presented in the study may lead to further investigation in the mechanism of asprin in cancer biology. However, the data presented in the study is bioinformatic and needs further validation through experimental biology.
2. The data is statistically fine and conclusions of the study are well stated.
3. The data needs more experimental validation.

Additional comments

1. Asprin has show to have some affects on melanoma and ovarian cancer. Did the authors have any data on these cancers? If so, please add data for the same.
2.the authors should describe more about the role of asprin in cancer metastasis.

---

## Round 0.2 · Minor Revisions

One of the reviewers indicated that that your manuscript contains some linguistic issues and therefore English needs to be copyedited by a colleague who is proficient in English and familiar with the subject matter, or a professional editing service. Please follow this advise and edit your manuscript accordingly.

Reviewer 1 ·

Basic reporting

improved

Experimental design

imroved

Validity of the findings

improved

Additional comments

improved

Reviewer 3 ·

Basic reporting

Sufficient
English needs to be copyedited by a native speaker

Experimental design

Sufficient

Validity of the findings

Sufficient

Additional comments

English needs to be copyedited by a native speaker.

---

## Round 0.3 · accepted · Accept

Thank you for addressing final critiques. The revised and edited version of this manuscript is accepted for publication now.

#